# Investigation of High Lift Force Generation of Dragonfly Wing by a Novel Advanced Mode in Hover

**Xiaohui Su [1,\*][ID], Kaixuan Zhang [1], Juan Zheng [1], Yong Zhao [2,\*], Ruiqi Han [3] and Jiantao Zhang [1]**

[1]   School of Hydraulic Engineering, Dalian University of Technology, Dalian 116024, China;
      zkx@mail.dlut.edu.cn (K.Z.); Zhengjuan@mail.dlut.edu.cn (J.Z.); zhangjt@dlut.edu.cn (J.Z.)
[2]   College of Engineering, Nazarbayev University, 53 Kabanbaybatyr Ave., Astana 010000, Kazakhstan
[3]   Department of Mechanical & Industrial Engineering, University of Toronto, 5 King's College Rd, Toronto,
      ON M5S 3G8, Canada; ruiqi.han@mail.utoronto.ca
\*   Correspondence: sxh@dlut.edu.cn (S.X.); yong.zhao@nu.edu.kz (Z.Y.)

**Abstract:** In the paper, a novel flapping mode is presented that can generate high lift force by a dragonfly wing in hover. The new mode, named partial advanced mode (PAM), starts pitching earlier than symmetric rotation during the downstroke cycle of the hovering motion. As a result, high lift force can be generated due to rapid pitching coupling with high flapping velocity in the stroke plane. Aerodynamic performance of the new mode is investigated thoroughly using numerical simulation. The results obtained show that the period-averaged lift coefficient, $C_L$, increases up to 16% compared with that of the traditional symmetrical mode when an earlier pitching time is set to 8% of the flapping period. The reason for the high lift force generation mechanism is explained in detail using not only force investigation, but also by analyzing vortices produced around the wing. The proposed PAM is believed to lengthen the dynamic stall mechanism and enhance the LEV generated during the downstroke. The improvement of lift force could be considered as a result of a combination of the dynamic stall mechanism and rapid pitch mechanism. Finally, the energy expenditure of the new mode is also analyzed.

**Keywords:** dragonfly wing; high lift force generation; partial advanced motion; vortex dynamics; hovering motion

## 1. Introduction

In the last two decades, attention has increasingly been paid to the aerodynamic performance of insect wings [1–5] due to the rising popularity of micro air vehicles (MAVs) [6,7]. Insects were the first creatures to develop flapping flight capability and remain in many ways unsurpassed in aerodynamic performance and maneuverability. Among insects, super flying capabilities, such as taking off backwards, flying sideways and landing upside-down et al. have been already discovered and reported [2]. People are very interested in the high-lift generation mechanisms and the corresponding energy expenditure in insect flight, mainly for the following two reasons. One is that biologists want to understand the effects of aerodynamic force production and energy expenditure from the point of view of physiology, ecology, evolution and other aspects of insects. The other is that engineers, who wish to develop small autonomous flying machines, want to understand the novel aerodynamics of flying insects and are eager to find inspiration from them [5].

Since the conventional aerodynamic theory of flapping wings has not been successful [3], much effort and progress has been made in revealing the dynamic mechanisms of insect flights. By using unsteady aerodynamic theory, lift force generation mechanisms have been identified and studied continually, such as clapping and fling [8], dynamic stall (delayed stall) [1], rapid pitch [2], wake

capture [2] and tip vortex (TipV) [9] as well as induced jet [10] for rigid wings. As for deformable wings, the effect of deformation has been investigated and it was concluded that deformation enhances lift force during both wake capture and delayed stall mechanisms [11]. The deformation of wings, including camber and angle of incidence, changes leading edge vortex (LEV) and trailing edge vortex (TEV) generation and development processes [11,12]. Among the above research works, most studies are carried out focusing on hovering motion. Hovering flight is a kind of flight mode where the body is assumed to be fixed in space and the freestream velocity is zero [11]. It is the most energetically expensive form of flight [8] and exceptionally fine control is also needed to remain stationary [13]. The power produced by the flapping of insect wings should sustain the insect itself in the air. To design and build MAVs with the capability of hovering flight, the mechanism of aerodynamic lift force generation of hovering flight is important to consider and worthy of detailed investigation.

In the numerical investigation of the flapping motion of insect wings, hovering action is generally simplified as a combination of two motions, translation and rotation. Normally, harmonic function is used to mimic rotation for all insect species, while there are two kinds of functions, trapezoidal function [2] and harmonic function [3], used to mimic translation depending on different insect species. Compared to simple harmonic function, trapezoidal function exaggerates the acceleration at the beginning and the deceleration near the end of a half-stroke and also makes the deceleration start at a later time (Figure 1). This pattern of wing translation, combined with proper wing rotation timing, can create aerodynamic forces by all the mechanisms currently known on wings. Different timings of wing rotation for the wings can be employed to provide control of forces on the wings during maneuver [1,2,5].

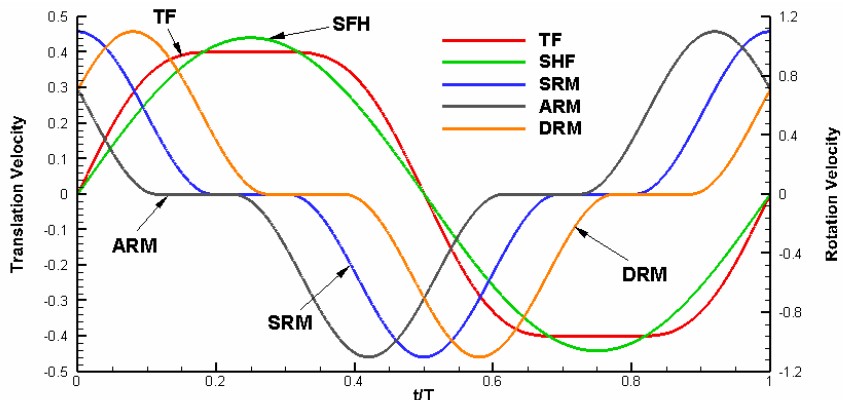

**Figure 1.** Non-dimensional translation (azimuthal rotation) velocity (left axis) and rotation velocity (right axis) in one cycle. (SHF: simple harmonic function; TF, trapezoidal function [5]; SRM: symmetrical rotation mode; ARM: advanced rotation mode; DRM: delayed rotation mode).

Currently, there are three combination modes appearing in the vast research works based on translation and rotation. When half of the wing rotation duration is completed at the end of a half-stroke and the other half in the beginning of the next half-stroke (Figure 1), the wing rotation is called symmetrical rotation mode (SRM); when the major part of rotation is finished before the stroke reversal, it is called advanced rotation mode (ARM); when the major part of rotation is only completed after the stroke reversal, it is called delayed rotation mode (DRM). The timing of the wing rotation can change the time history and the mean value of the aerodynamic forces significantly, especially in the case of translational velocity varying as a trapezoidal function [2,5].

Inspired by Dickinson's work [2], investigation of the high lift force generation of dragonfly wings by changing rotation timing are carried out in this study. During the investigation of the ARM coupling with harmonic translation function, good improvement of lift force during downstroke is found, but with poor performance of lift force during upstroke if ARM is also adopted near the end of the upstroke. The negative performance of lift force during upstroke action is even worse

compared with the results of SRM. As a result, the period-averaged lift force does not improve much or even worse. Therefore, it is proposed that the ARM be adopted for the downstroke only for its advantage, while the SRM is used for the upstroke, which can be named as partial advanced mode (PAM). Hereby, in this paper, we carry out numerical investigations on the aerodynamic performance of dragonfly wings using the proposed PAM. An unstructured mesh unsteady incompressible 3D flow solver, named TetraALEFSI, which was originally developed by Zhao's group [14–16] and was recently enhanced by Su et al. with an arbitrary Lagrangian-Eulerian (ALE) function [12,17], is used for the numerical simulations.

The structure of this paper is as follows: the governing equations of the unsteady incompressible 3D ALE flow solver are briefly introduced in Section 2. In Section 3, the geometry and kinematics of the dragonfly wing during hovering motion is described. In Section 4, high lift force generation mechanisms due to the proposed partial advanced mode are investigated and discussed. Conclusions are made in the last section.

## 2. Mathematical and Numerical Formulation

The three-dimensional incompressible unsteady Navier-Stokes governing equations, modified by the artificial compressibility method (ACM) with dual time steps and arbitrary Langrangian-Eulerian (ALE) method in non-dimensional vector form, are as follows:

$$\mathbf{C}\frac{\partial \mathbf{W}}{\partial \tau} + \mathbf{K}\frac{\partial \mathbf{W}}{\partial t} + \nabla \cdot \mathbf{F_c} = \nabla \cdot \mathbf{F_v} \tag{1}$$

where,

$$\mathbf{W} = \begin{bmatrix} p \\ u \\ v \\ w \end{bmatrix} \mathbf{F}_c = \begin{bmatrix} \mathbf{U} \\ u\mathbf{U} + p\delta_{ij} \\ v\mathbf{U} + p\delta_{ij} \\ w\mathbf{U} + p\delta_{ij} \end{bmatrix} \mathbf{F_v} = \begin{bmatrix} 0 \\ (1/\mathrm{Re}) \cdot \nabla \cdot u \\ (1/\mathrm{Re}) \cdot \nabla \cdot v \\ (1/\mathrm{Re}) \cdot \nabla \cdot w \end{bmatrix}$$

$$\mathbf{K} = \begin{bmatrix} 0 & 0 & 0 & 0 \\ 0 & 1 & 0 & 0 \\ 0 & 0 & 1 & 0 \\ 0 & 0 & 0 & 1 \end{bmatrix} \mathbf{C} = \begin{bmatrix} \frac{1}{\beta} & 0 & 0 & 0 \\ 0 & 1 & 0 & 0 \\ 0 & 0 & 1 & 0 \\ 0 & 0 & 0 & 1 \end{bmatrix}$$

where $\mathbf{W}$ is the vector of dependent variables and $\mathbf{F}_c$ and $\mathbf{F_v}$ are the convective flux and viscous flux vectors, respectively. $\beta$ is a constant called artificial compressibility whose value affects the solution convergence to steady state. $\mathbf{K}$ is the unit matrix with first element equal to zero and $\mathbf{C}$ a preconditioning matrix. $\mathbf{U} = \mathbf{U}_f - \mathbf{U}_m$ is the velocity vector, $\mathbf{U}_f$ and $\mathbf{U}_m$ are the fluid velocity and grid velocity, respectively; *Re* is the Reynolds number.

The governing equation, Equation (1) is discretized on an unstructured tetrahedral grid using the control volume method. The details of discretization of the governing equation, numerical validations of the proposed solver, as well as the moving mesh algorithm can be found in reference [12,14–17].

## 3. Geometry and Kinematics of Dragonfly Wing

### 3.1. Geometry of Dragonfly Wing

The hindwing of a dragonfly, as used in this research, is described as a rigid plate with a span length of *R*, 4.6 cm, average chord length of *c* = 1.12 cm, and a thickness of 1% of average chord length, about 0.0112 cm. The symmetric plane is located on the left 0.61 cm away from the root of wing, and the rotation axis of the wing is 0.28 cm, about *c*/4 from the leading edge. The simplified geometric model of the hindwing of a dragonfly is plotted in Figure 2.

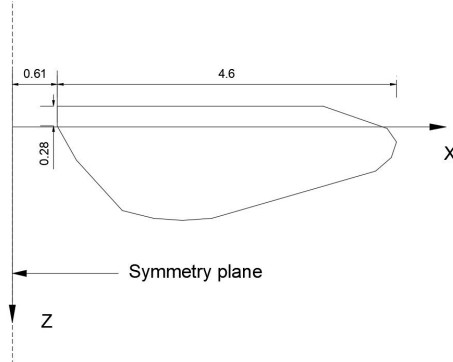

**Figure 2.** Simplified geometric model of the dragonfly hindwing.

### 3.2. The Traditional Kinematics of Hovering Motion

The traditional kinematics of hovering motion are described using formulas presented by Norberg [18] and Sun [19], and the sketch of the flapping motion is defined and presented in Figure 3. It assumes that the stroke plane, also named the flapping plane, has an inclined angle, β, about 52° to the horizontal plane. Thus, only two parameters, the flapping angle, Φ, which is relative to the *Y-Z* plane, and the angle of attack (AOA), α, which is relative to the flapping plane, will be used together to control the movement of the wing. The flapping frequency is set to 36 Hz.

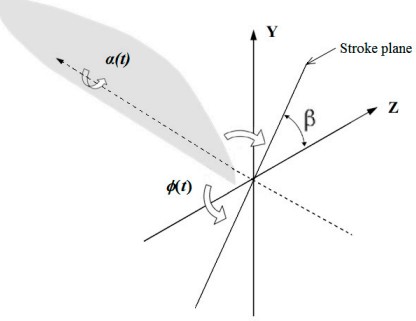

**Figure 3.** Sketch of the flapping motion of the wing.

The instantaneous flapping angle, Φ(t), and relative AOA angle, α(t), are described in Equations (2) and (3), respectively:

$$\varphi(t) = \varphi_0[\cos(2\pi f t) - 1] \tag{2}$$

$$\alpha(t) = \alpha_d - \frac{(\alpha_d - \alpha_u)}{\Delta t_r}\left[(\mathrm{mod}(t,T) + T - t_{ru}) - \frac{\Delta t_r}{2\pi}\sin(2\pi\frac{\mathrm{mod}(t,T) + T - t_{ru}}{\Delta t_r})\right]$$

$$0 \leq \mathrm{mod}(t,T) < \frac{\Delta t_r}{2}$$

$$\alpha(t) = \alpha_u$$

$$\frac{\Delta t_r}{2} \leq \mathrm{mod}(t,T) < t_{rd}$$

$$\alpha(t) = \alpha_u + \frac{(\alpha_d - \alpha_u)}{\Delta t_r}\left[(\mathrm{mod}(t,T) - t_{rd}) - \frac{\Delta t_r}{2\pi}\sin(2\pi\frac{\mathrm{mod}(t,T) - t_{rd}}{\Delta t_r})\right]$$

$$t_{rd} \leq \mathrm{mod}(t,T) < t_{rd} + \Delta t_r$$

$$\alpha(t) = \alpha_d$$

$$t_{rd} + \Delta t_r \leq \mathrm{mod}(t,T) < t_{ru}$$

$$\alpha(t) = \alpha_d - \frac{(\alpha_d - \alpha_u)}{\Delta t_r}\left[(\mathrm{mod}(t,T) - t_{ru}) - \frac{\Delta t_r}{2\pi}\sin(2\pi\frac{\mathrm{mod}(t,T) - t_{ru}}{\Delta t_r})\right]$$

$$t_{ru} \le \mathrm{mod}(t,T) < t_{ru} + \frac{\Delta t_r}{2} \tag{3}$$

where $\Phi_0$ is the amplitude of the flapping angle and $\varphi_0 = 0.602$, according to reference [19]. $\alpha_d = 66°$ and $\alpha_u = 14°$, $\Delta t_r$ is the duration of the wing flip and $t_{ru}$ and $t_{rd}$ are time moments that the wing starts to rotate in the upstroke and downstroke actions. According to reference [19], $\Delta t_r = 0.39T$ is the same for both the downstroke and upstroke in SRM, and $t_{rd} = 0.305T$ and $t_{ru} = 0.805T$, where $T$ is the time period of AOA movement. Besides SRM, there are another two unsymmetrical modes, called ARM and DRM, respectively. These two unsymmetrical modes are obtained by shifting $\alpha(t)$ with a negative angle for the ARM and with a positive angle for the DRM. Both ARM and DRM can be expressed by Equation (4).

$$\alpha(t) = \alpha(t)_s \mp \alpha_0 \tag{4}$$

where subscript $s$ denotes AOA in SRM. In fact, the physical meaning of two unsymmetrical modes is to shift wing rotation timing in downstroke and upstroke. The time histories of the flapping angle and angle of attack in two cycles for the above mentioned three modes are shown in Figure 4. It is noted that when the relative AOA is zero, the wing is actually $40°$ relative to the flapping plane.

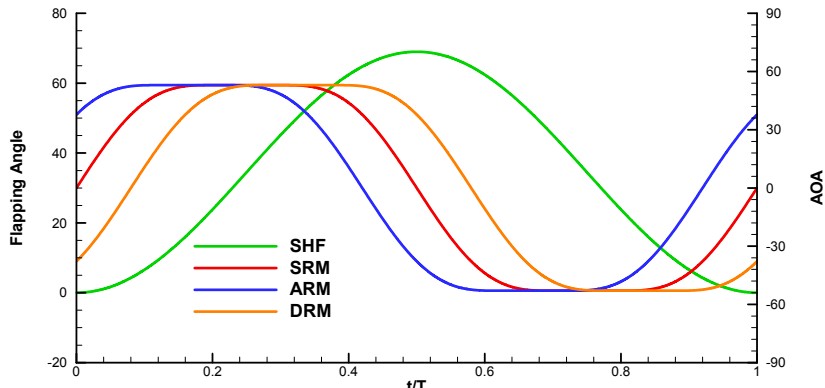

**Figure 4.** The time histories of flapping (green) and relative AOA angles in one period during hovering motion. (SHF: simple harmonic function; AOA: angle of attack; SRM: symmetrical rotation mode; ARM: advanced rotation mode; DRM: delayed rotation mode. Units of flapping angle and relative AOA are degree).

### 3.3. The New Kinematics of Flapping Motion

In this paper, a partial advanced mode (PAM) is proposed by shifting the wing rotation timing for near the end of the downstroke while keeping the wing rotation in upstroke motion intact, as that of the symmetrical mode. The wing downstroke rotation is now given as $t_{rd} = (0.305 \mp t_0)T$, where $t_0$ is the ratio of shifted start time to time period. A new rotation kinematics of wing flapping motion is generated, meanwhile the time histories of the flapping angle and the angle of attack in two cycles are shown in Figure 5.

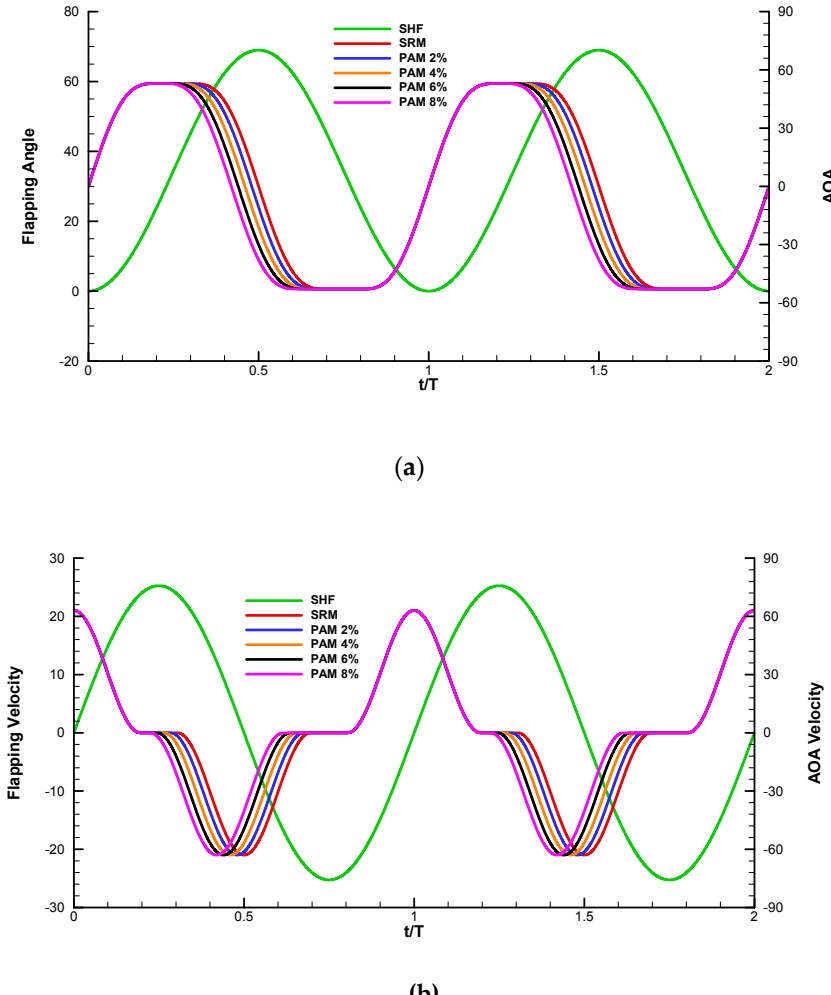

**Figure 5.** The time histories of flapping and attack angles (**a**) and flapping and AOA angular velocities (**b**) in two periods during hovering motion. (SHF: simple harmonic function; AOA: angle of attack; SRM: symmetrical rotation mode; PAM: partial advanced mode). (**a**) The time histories of flapping (green) and AOA angles; (**b**) The time histories of flapping (green) and AOA angular velocities.

### 3.4. Model Setup

In the model, air density, $\rho = 1.185 \text{ kg/m}^3$, and dynamic viscosity, $\mu = 1.831 \times 10^{-5} \text{ kg} \cdot \text{m}^{-1} \cdot \text{s}^{-1}$, are set in a still air. Transient analysis calculates 30 cycles to ensure the final solution in one period to attain convergence status. The time step is set to $5 \times 10^{-6}$ s. The computational domain is chosen as a sphere with a radius of 30 cm, which is more than six times the wingspan of 4.6 cm. The center of the sphere is located at the point of origin of the coordinate system *X*, *Y* and *Z*, which can be seen in Figures 2 and 3. There are two types of boundaries in the above computational domain, wing surface and sphere surface. In the paper, the wing surface is a moving boundary, while the sphere surface is the far field boundary. The far field and moving boundary conditions are selected as opening and non-slip wall boundary conditions respectively.

## 4. Results and Discussions

### 4.1. Mesh Convergence Test

There are four meshes used to verify mesh convergence. The total mesh cells of the model are 35,566, 68,497, 11,3614 and 23,3150, which are summarized in Table 1.

**Table 1.** Mesh information.

| Mesh Model | Wing Surface Mesh | Total Mesh | Node Point |
| --- | --- | --- | --- |
| Coarse | 1024 | 35,566 | 6566 |
| Medium | 1024 | 68,497 | 12,058 |
| Fine | 2786 | 113,614 | 19,995 |
| Finer | 6084 | 233,150 | 40,746 |

As an example, the mesh of the whole computational domain (a) and the cross section and zoom-out view of the mesh around the wing (b and c) are presented in Figure 6. The instantaneous pressure values of those monitoring points are shown in Figure 7.

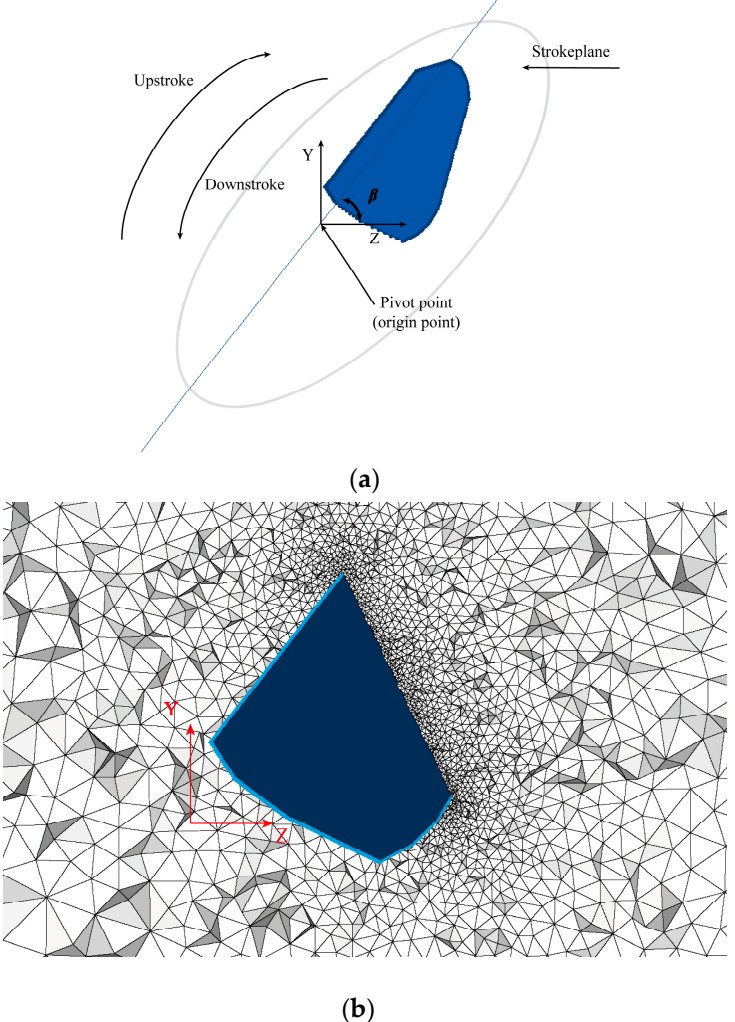

(**a**)

(**b**)

**Figure 6.** Mesh grid (fine) and motion parameters for the simulation of a dragonfly hindwing during hovering motion in the initial time step. (**a**) Flapping motion parameters in the initial time step; (**b**) grid of computational flow field in the initial time step.

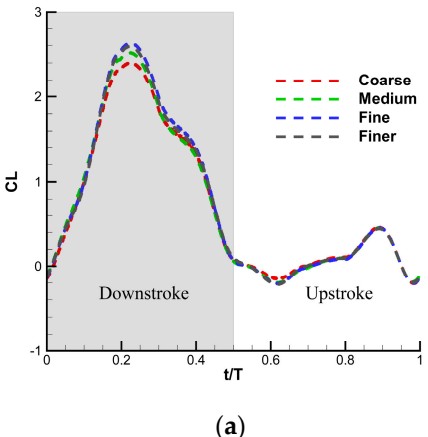
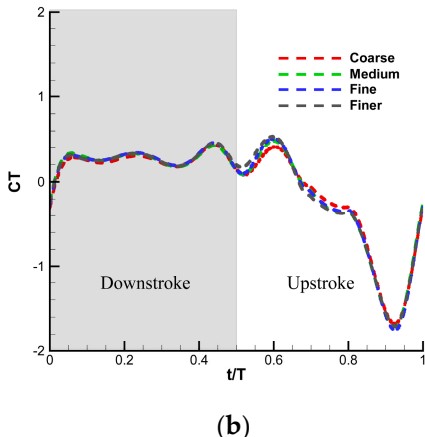

(**a**)          (**b**)

**Figure 7.** A comparison of the instantaneous lift and thrust coefficients of the wing in the hovering motion between coarse, medium, fine and finer grids (**a**) four kinds of lift force coefficient curves, $C_L$, in comparison; (**b**) four kinds of thrust coefficient curves, $C_T$, in comparison.

Figure 7 shows that with the increasing of mesh density, the instantaneous lift and thrust coefficients gradually converge to stable and mesh independent values. When the mesh used is the fine grid, the relative error of the instantaneous values is below 1%. Therefore, the solution with the fine mesh could be considered as reaching mesh convergence. Taking into account the requirements of computational accuracy and time cost, the fine mesh is used as the optimal computational mesh for subsequent studies.

*4.2. Aerodynamic Performance of Symmetrical Model and Validation of Results*

In the study of insect flapping flight, aerodynamic factors, including the lift force coefficient, $C_L$, and thrust coefficient, $C_T$, are the most important in order to characterize the aerodynamic forces. In this study, $C_L$ denotes the aerodynamic component perpendicular to the horizontal plane pointing to the y-axis for lifting the weight of the dragonfly, while the $C_T$ represents the aerodynamic component orthogonal to the $C_L$ directed to the z-axis. The $C_L$ and $C_T$ are obtained by Equation (5) and (6) as follows.

$$C_L = \frac{F_Y}{0.5\rho U^2 S_h} \tag{5}$$

$$C_T = \frac{F_Z}{0.5\rho U^2 S_h} \tag{6}$$

where $U$ = 4.0724 m/s is the dimensionless speed, and $S_h$ is the hindwing area of 5.152 cm$^2$.

The $C_L$ and $C_T$ obtained by the current model are presented in Figure 8, where the results obtained by Sun [19] are also plotted for comparison and validation purposes. The period-averaged $C_L$ of the present model is 0.736, while the value using Sun's model [19] is 0.675, which shows a relative error of about 9%. The lift force generation mechanisms including wake capture and dynamic stall, as well as rapid pitch, are clearly captured, which serves to validate the capability of the current numerical model.

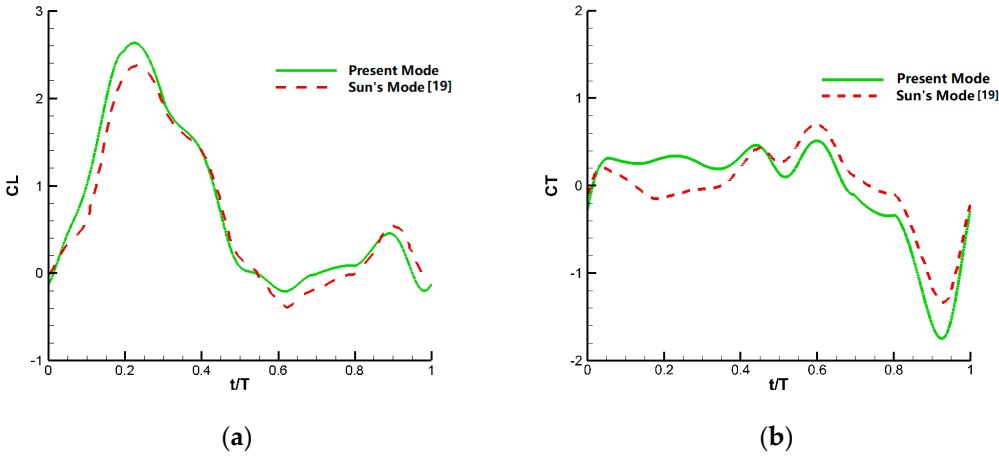

**Figure 8.** The comparison of the profiles of $C_L$ and $C_T$ in one hovering motion cycle. (**a**) The time history of $C_L$; (**b**) the time history of $C_T$.

### 4.3. Aerodynamic Performance of PAM

In this section, aerodynamic performance of the proposed PAM is investigated. The numerical investigations are conducted in the following three steps. Firstly, the shift time, $t_0$, effects of PAM are investigated. Secondly, with the typical shift time value, $t_0$, from the first step, the results of the SRM, ARM and PAM are compared. Thirdly, analyses and discussions of the vortex formations and energy consumption are carried out in order to provide a proper aerodynamic explanation for the results obtained and to further evaluate the performance of the proposed PAM.

### 4.3.1. The Shift Time Effects of PAM and PDM

As described in Figure 5, the PAM cases, with a set of shift times, $t_0$, such as 2%T, 4%T, 6%T and 8%T, are implemented. Aerodynamic performances of the above cases are simulated using our inhouse solver TetraALEFSI [12,14,17], and the results are plotted in Figure 9.

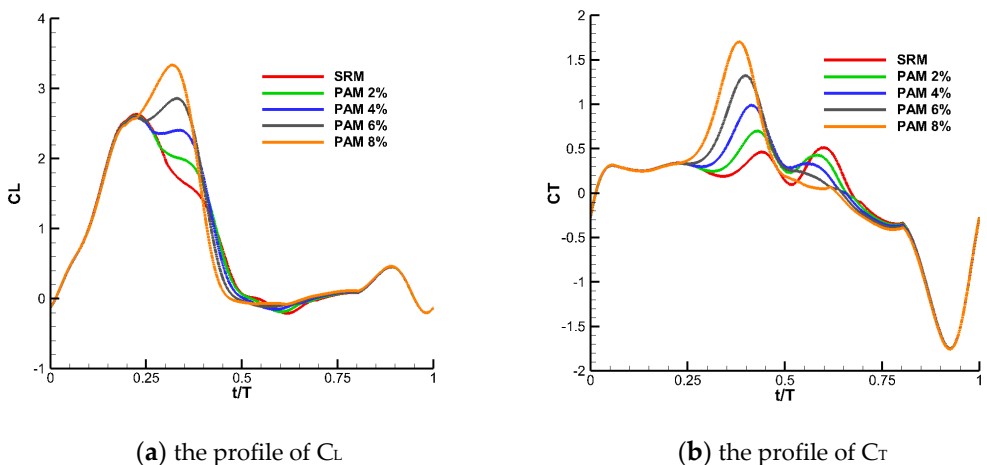

(**a**) the profile of $C_L$　　　　　　　　　　　　　　　　(**b**) the profile of $C_T$

**Figure 9.** Time histories of $C_L$ and $C_T$ for PAM with different shift time values.

Figure 9 reveals that aerodynamic forces are improved dramatically by using the PAM in the hovering motion. Compared with the results of the SRM, significant improvement of lift force appeared during the partial advanced shifting phase, 0.22T~0.41T. According to the results, the higher the shift time value, the larger the lift force that could be obtained. When the shift time reaches 8%T in PAM, the maximum lift force is about 3.4, nearly 1.36 times larger than the maximum value of the SRM, 2.5. At the same time, different from $C_L$ distribution in SRM, lift force resulting from the PAM continues

increasing in the cases of 6%T and 8%T. The time moment for the maximum lift force appearing is shifted later for the 6%T and 8%T cases, for example, 0.32T for the 8%T case, almost 0.07T delayed from the time moment in the SRM case. Together with the increase of the lift force, the thrust force is higher during the partial shifting phase. The maximum value of the thrust force is about 1.78 for the 8%T case. The aerodynamic force during upstroke actually has no change since the same flapping action as the one in SRM is used.

### 4.3.2. The Comparisons of PAM and ARM Modes

To further understand the effects of the proposed PAM on lift force generation, systematical comparisons with PAM and ARM modes are carried out. The shift time value, 8%T, is chosen for the PAM and ARM due to the outstanding performance obtained in the previous section.

Figure 10 shows the time histories of $C_L$ and $C_T$ during one cycle of hovering motion using the above-mentioned three modes. The comparison is mainly focused on aerodynamic forces from the PAM and ARM. Significant difference has been found in the upstroke. Rapid pitch in the first half of the upstroke in ARM does not play any positive effect on lift force generation. On the contrary, both the ARM lift and thrust forces decrease rapidly in the upstroke due to the orientation of the wing at the ends of both strokes of $40°$ relative to the flapping plane in the downward direction. Only during the second half of the upstroke, the lift force in ARM increases slightly, also due to the rapid pitch, but being much less effective than the corresponding downstroke because of the above-mentioned wing orientation relative to the flapping plane. Therefore, one can conclude that the overall lift force could only be improved by using PAM alone.

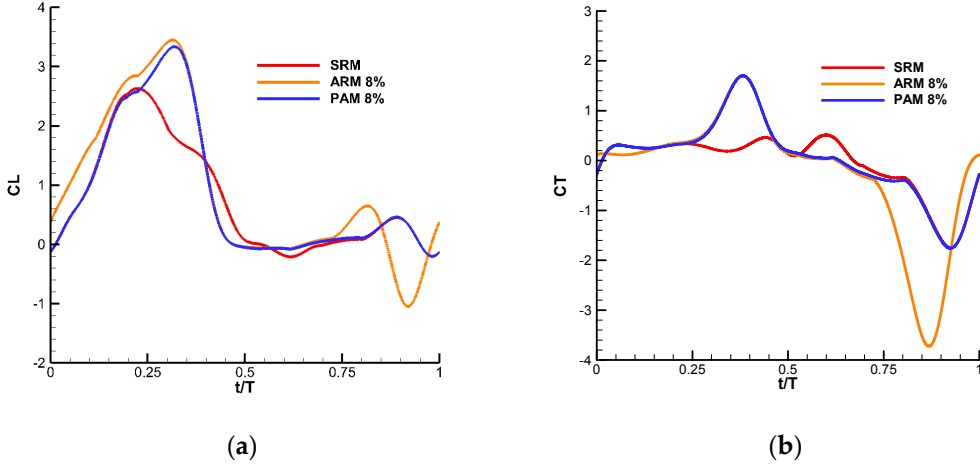

**Figure 10.** Time histories of $C_L$ and $C_T$ during one cycle of hovering motion for three flapping modes. (**a**) The profile of $C_L$; (**b**) the profile of $C_T$.

### 4.4. Vortex Analyses in Lift Force Generation by PAM

In Section 4.3, aerodynamic forces from PAM are described and the significant points of the proposed mode have been obtained mainly based on the $C_L$ and $C_T$ profiles and the comparisons between SRM and ARM. In this section, further investigation on the lift force generation mechanism in PAM is to be carried out with focus on vortex analyses.

Figure 11 shows snapshots of typical fluid structures, in the form of pressure contours, streamlines and iso-vorticity surfaces in one cycle of hovering motion with the PAM. The cross-sectional plane is chosen with 0.65R for pressure contours and streamlines, while the contour values of the iso-vorticity surface are set to $-18,400s^{-2}$. The background color (lilac) of the vorticity contour is the cross-sectional plane which intersects with the iso-vorticity surfaces (grey) in Figure 11. Figure 12 plots the lift force time history for the case in Figure 11 with the same time instants marked by solid points and their corresponding iso-vorticity surfaces.

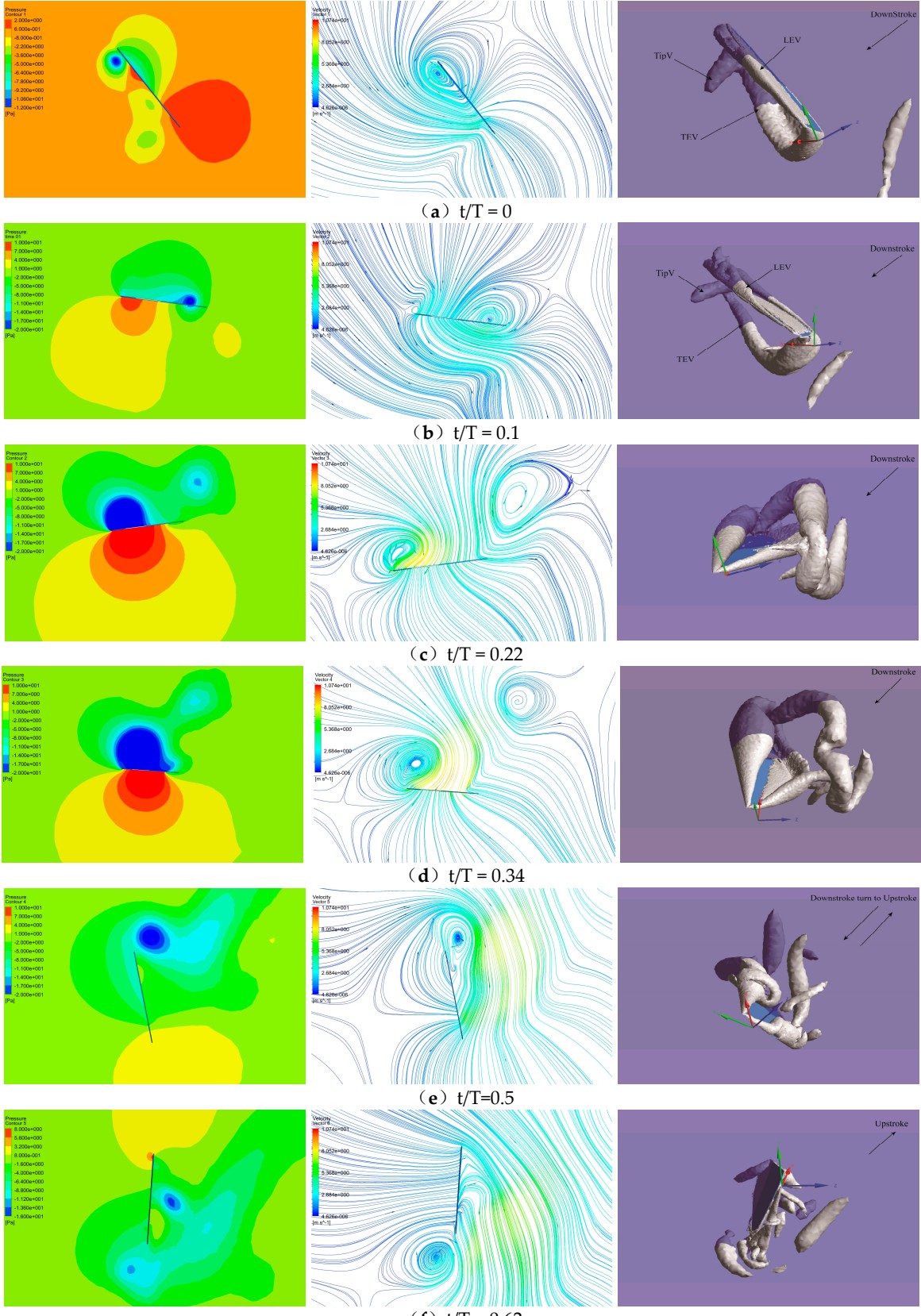

（**a**）t/T = 0

（**b**）t/T = 0.1

（**c**）t/T = 0.22

（**d**）t/T = 0.34

（**e**）t/T=0.5

（**f**）t/T = 0.62

**Figure 11.** *Cont.*

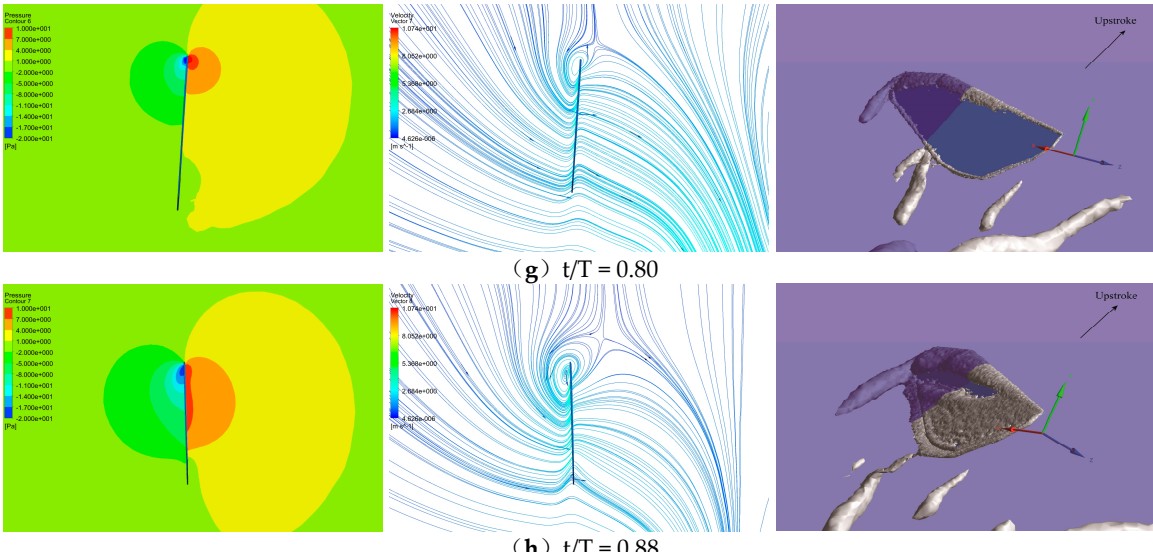

(**g**) t/T = 0.80

(**h**) t/T = 0.88

**Figure 11.** Pressure contours (left), streamlines (middle) and iso-vorticity surfaces (right). At different time instants with PAM 8%T in one cycle of hovering motion (**a–h** corresponds to t/T = 0, 0.1, 0.22, 0.34, 0.5, 0.62, 0.80 and 0.88, respectively).

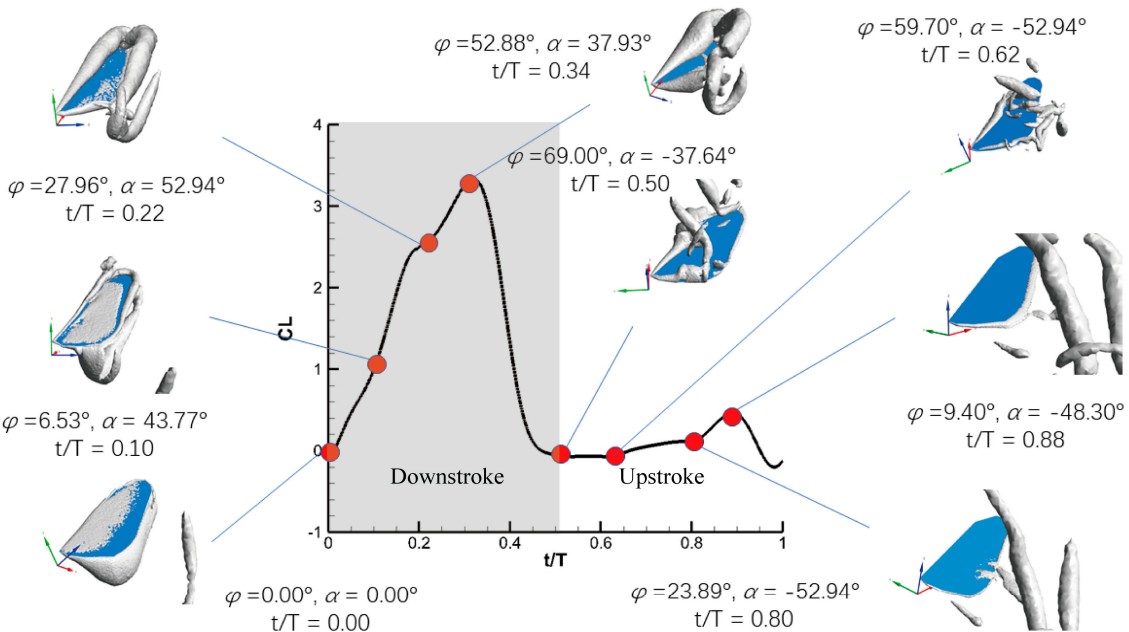

**Figure 12.** The lift force and iso-vorticity surface plots for 8%T PAM.

At time instant t/T = 0, unshed vortices in the previous cycle still exist at the edge of the wing. At this moment, the wing is starting to accelerate downward, accompanied by a counterclockwise rotation.

When t/T = 0.1, air moves around the wing from the leading edge to the trailing edge and around the wing tip, and a new vortex system around the wing is generated which is composed of a leading edge vortex (LEV), a trailing edge vortex (TEV) and a tip vortex (TipV). In fact, the new vortex system has been found as a whole at the beginning, as the vortices, such as LEV, TEV and TipV, are well established from the beginning. It is noted that the LEV is a clockwise vortex while the TEV is counterclockwise. At the same time, air impacts the lower surface of the wing and goes though the wing, which enhances the LEV, TEV and TipV. A large negative pressure area is formed on the upper surface of the wing, while a positive pressure area is formed on the lower surface. The pressure difference between the upper and lower surfaces generates a vertical upward lift. It can be seen from

the iso-vortex surfaces that the vortex on the edge of the wing in the previous cycle gradually falls off, which shows that the mechanism of wake capture is almost over and the dynamic stall mechanism will start to work.

When t/T = 0.22, lift force on the wing reaches an instantaneous peak due to the dynamic stall mechanism. From the diagrams shown in Figure 11c, it can be seen that the clockwise vortex at the leading edge is still attached to the upper surface of the wing, but the counterclockwise vortex at the trailing edge starts to fall off, causing the pressure difference between the upper and lower surfaces to decrease. Thus, the lift force will decrease if there is no more other action. However, the PAM can now be activated and play a role at this moment, which will start rapid pitch in the second half of the downstroke.

When t/T = 0.34, the increasing lift force due to PAM reaches its peak, which is also the maximum lift force generated in the whole cycle. From 0.22T to 0.34T, the wing partial advanced rotation in a clockwise direction leads to the LEV continuing development and keeps it attached to the leading edge. The action of the proposed PAM indeed lengthens the dynamic stall and enhances the lift force.

At t/T = 0.5, the LEV rotates clockwise to the right and falls off the wing eventually, which means that the wing cannot produce lift from this LEV anymore. At this point, the wing starts to accelerate up on the right.

From t/T = 0.62 to t/T = 0.88, the wing moves upward along the stroke plane with a small AOA angle, and the air flow around the wing is almost unimpeded, thus no lift force is generated. The vortex ring is basically completely detached from the wing.

In conclusion, at the beginning of the downstroke, the wing accelerates downward and the LEV starts to take shape, which leads to an increase in lift force. In the intermediate translational stage, the LEV is further enhanced resulting in instantaneous lift peak due to the dynamic stall mechanism. Both PAM and SRM share the same motions, therefore their mechanisms of lift generation are basically the same up to a quarter of the cycles. After this, the wing starts to flap with greater rotational speed using the PAM, thus the lift can continue to increase. However, the rotational speed of the wing has been reduced to a very small value for the SRM, thus the lift force cannot be increased. This is a new way to generate lift force using a combination of dynamic stall and rapid pitch mechanisms. Moreover, PAM follows the same motion as SRM during upstroke since lift force cannot be enhanced significantly using ARM (Figure 10). In fact, this is the reason that the partial advanced mode is proposed because the lift force generated during the upstroke in the ARM is very poor.

### 4.5. Energy Consumption Analyses Using PAM

The aerodynamic parameters, including time period-averaged $C_L$, $C_T$ and power coefficient $C_P$, as well as figure-of-merit M, are calculated and summarized in Table 2 and plotted in Figure 13. Figure 14 shows the time-averaged aerodynamic parameters using the PAM.

**Table 2.** Time-averaged aerodynamic parameters for partial advanced mode.

| Model | $\overline{C_L}$ | $\overline{C_T}$ | $\overline{C_P}$ | $\eta$ |
|---|---|---|---|---|
| Symmetry | 0.7374 | −0.0593 | 0.196 | 3.7622 |
| 2% | 0.7642 | −0.0368 | 0.214 | 3.5710 |
| 4% | 0.7946 | −0.0124 | 0.242 | 3.2835 |
| 6% | 0.8240 | 0.0134 | 0.275 | 2.9964 |
| 8% | 0.8522 | 0.0401 | 0.312 | 2.7314 |

Energy consumption of the wing during one flapping period is calculated using the following formulations:

$$P = \int_0^T \oint_A (\vec{U}(t) \cdot \vec{n}(t))p(t)dA(t)d(t) \tag{7}$$

where $P$ is the power supporting wing movement during one hovering motion cycle. The efficiency of power, $C_P$, is calculated by

$$C_P = \frac{P}{0.5\rho U^3 S_h} \tag{8}$$

where $\rho$ is the density of air, $\rho = 1.185 \text{ kg/m}^3$, $S_h$ is the area of the hindwing, $S_h = cR = 1.12 \times 4.6 = 5.152 \text{ cm}^2$, and $U$ is the reference velocity, calculated at the location of 0.65R, $U = 4.0724 \text{ m/s}$.

The wing efficiency is evaluated by a parameter, $\eta$, the lift-to-power ratio [20], which is defined as follows:

$$\eta = \frac{\overline{C_L}}{\overline{C_p}} \tag{9}$$

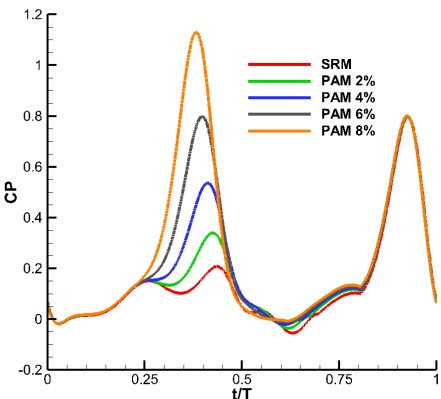

**Figure 13.** The time histories of power coefficient $C_P$ for partial advanced mode.

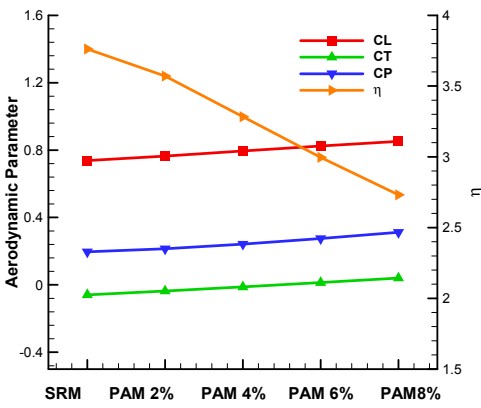

**Figure 14.** Time-averaged aerodynamic parameters for different start times of stroke reversal.

Table 2 shows the time-averaged $C_L$, $C_T$, $C_P$ and $\eta$ in PAM with different shift times. Compared with $C_L = 0.7374$ using ARM, PAM has a time-averaged $C_L = 0.8522$, about a 16% increase in lift force in the 8%T PAM case. Lift force is dramatically improved by the PAM. The time-averaged $C_T$ also increases with increasing shift time in PAM. The time-averaged $C_P$ reaches 0.312 in the 8%T PAM case, 0.116 larger than that of the SRM case, which means that although the proposed partial advanced mode could improve lift force, energy consumption has also increased and the corresponding wing efficiency parameter has decreased significantly.

## 5. Conclusions

In the paper, a numerical method is presented, validated and then applied to numerically investigate life force generation mechanisms of a dragonfly wing during hovering motions. High lift force is generated by implementing the partial advanced flapping mode during hovering motion,

namely PAM. The aerodynamic performance of the proposed PAM is numerically investigated and compared with those of the traditional SRM, as well as the ARM. The results show that lift forces, not only the peak value but also the period-averaged values, are dramatically improved by using PAM. Peak lift force increases by 1.36 times, while period-averaged Cl increases by 16% in the 8%T PAM case, compared with those of the SRM. The high lift force generation mechanism is analyzed and investigated by studying flow structures, including pressures and streamlines as well as iso-vorticty surfaces, around the wing. The proposed PAM is believed to lengthen the dynamic stall mechanism and enhance the LEV generated during downstroke. The improvement of lift force could be considered as a result of a combination of the dynamic stall mechanism and rapid pitch mechanism. However, it should be noted that although the PAM mode could improve lift force, the increase of energy consumption and corresponding decrease in wing efficiency parameter have to be considered as byproducts of this mode.

**Author Contributions:** Conceptualization, X.S. and K.Z.; methodology, K.Z., J.Z. (Juan Zheng) and J.Z. (Jiantao Zhang); software, K.Z. and R.H.; validation, K.Z., J.Z. (Juan Zheng) and R.H.; formal analysis, J.Z. (Jiantao Zhang); investigation, K.Z. and J.Z. (Juan Zheng); resources, J.Z. (Jiantao Zhang); data curation, K.Z. and J.Z. (Juan Zheng); writing—original draft preparation, X.S., K.Z.; writing—review and editing, X.S., K.Z. and Y.Z.; visualization, K.Z. and J.Z. (Juan Zheng); supervision, X.S.; project administration, X.S.; funding acquisition, X.S. All authors have read and agreed to the published version of the manuscript.

**Funding:** This research is supported by The National Natural Science Foundation, project No: 11672059. The financial support is gratefully acknowledged.

**Conflicts of Interest:** The authors declare no conflict of interest.

## Nomenclature

| symbol | meaning | unit |
| --- | --- | --- |
| $c$ | average chord length | cm |
| C | preconditioning matrix | - |
| $C_L$ | lift force coefficient | - |
| $C_T$ | thrust coefficient | - |
| $\mathbf{F_c}$ | convective flux | - |
| $\mathbf{F_v}$ | viscous flux | - |
| **K** | unit matrix with first element equal to zero | - |
| $R$ | wing length | cm |
| $Re$ | Reynolds number | - |
| $t$ | time | s |
| $t_0$ | ratio of shifted start time to time period | s |
| $t_{rd}$ | divided point from translate to rotate in downstroke | s |
| $t_{ud}$ | divided point from translate to rotate in upstroke | s |
| $\Delta t_r$ | duration of wing flip | s |
| $T$ | flapping period | s |
| X,Y,Z | rectangular coordinate system | - |
| $\alpha$ | angle of attack | $^0$ |
| $\alpha_d$ | midstroke geometric angle of attack of downstroke | $^0$ |
| $\alpha_u$ | midstroke geometric angle of attack of upstroke | $^0$ |
| $\alpha_0$ | phase angle of flapping motion | $^0$ |
| $\alpha_s$ | angle of attack in symmetrical rotation mode | $^0$ |
| $\beta$ | artificial compressibility coefficient | - |
| $\beta$ | inclined angle | $^0$ |
| $\eta$ | wing efficiency parameter | - |
| $\mu$ | dynamic viscosity | kg/m/s |
| $\rho$ | air density | kg/m$^3$ |
| $\varphi$ | flapping angle | $^0$ |
| $\varphi_0$ | amplitude of flapping angle | $^0$ |

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
