# Peer review of "Investigation of High Lift Force Generation of Dragonfly Wing by a Novel Advanced Mode in Hover"

_fluids, doi:10.3390/fluids5020059_

Round 1
Reviewer 1 Report
Reviewer's comments on Manuscript ID: fluids-773692
Title: Investigation of High Lift Force Generation of 3D Dragonfly Wing by Partial Advanced Mode during Hovering Motions
This research investigated flapping mode to generate high lift force by dragonfly wing. The research has been prepared well but requires few points to be revised before consideration for publication as follows:
- This is a research paper not a review paper, therefore I believe the relevant & updated references must be used only. In addition there is a mistake at Reference section for references 49-52 which must be revised.
- Nomenclature must be added to ease the understanding of the text.
- The computational domain and BC’s should be defined properly
Reviewer 2 Report
This paper presents a partially adjusted wing pitching motion profile as a way of increase aerodynamic lift of a flapping wing. Based on the hindwing of a dragonfly, they investigated the effect of the timings of the wing rotation; this includes symmetrical, advanced, and delayed as Dickinson et al. (1999) had addressed, and the partially-changed timing of the wing rotation they addressed here. They showed that the CL can be enhanced up to 16% with the advanced timing of the wing rotation of 0.08T, but it also can reduce the efficiency. Overall, the paper presents original analysis and somewhat new contribution to the area of flapping-wing aerodynamics; I recommend to accept this manuscript after the major revisions. For the publication, all the specific points given below should be addressed. In addition, the Authors enlist a writing service to improve the manuscript stylistically, because more than some sentences have the wrong tenses and grammatical errors, frequently interrupting the focus.
1) Title. 3D looks redundant. "partial advanced mode" does not give any information. "during hovering motions" can be reduced to "in hover". Kindly revise the title in a concise manner.
2) Abstract, line 3. The partially advanced timing of the wing rotation here denotes the earlier pitching motion at the end of upstroke than “symmetric rotation”, not the “conventional advance mode”. Kindly revise it.
3) Abstract, line 6. Computer simulation looks weird. Can you change it with the other specific technical terminology?
4) Abstract, lines 8-10. The reason why should be explained herein Abstract. Kindly revise.
5) Introduction, paragraph 1, lines 1-2. Such a large number of references is completely not acceptable. Use the driving/review papers or books dealing with the flapping-wing aerodynamics, instead.
6) Introduction, paragraph 1, lines 1-2. 1) This can be misleading. Even recent studies on the aerodynamic model are based on the 'quasi-steady' concept and most of them show reasonable estimation capability. I think the quasi-steady state theory the authors state there is 'conventional aerodynamics'. Revise it with considering above. 2) What does 'reject' mean here?
7) Introduction, paragraph 1, line 6. The author called tip vortex as TipEV, but there is no 'E' in the tip vortex. Change this abbreviation.
8) Page 2, line 1. The sentence finishes with "~~~ delated stall mechanisms." need a reference.
9) Page 2, line 1. Nowadays the "Wing morphing" is indicating the change in the wing shape in an active way, not the deformation of the flapping wing. Kindly revise it.
10) Figures 4 and 5. I would like to use a degree, instead of a radian.
11) Figure 6. Much information is omitted. Stroke direction, stroke plane, pivot point, lift and drag directions, coordinate system, etc. Provide the kinematic and dynamic variables as much as you can.
12) Figure 7. I know where the down or upstroke is, but most of the readers would not recognize at once. Add the phase in the graph.
13) Page 9, 1st paragraph. Just a thought, 9% error is acceptable when it comes to the different wing shape; the authors do not have to explain with too much detail here.
14) Page 9, 4.3.1, 1st paragraph. TetraALEFSI definitely needs a reference.
15) Figure 11. This needs much more information such as the location of the pivot point and cross-section, stroke direction, etc. the color bar for the contour should be enlarged.
16) Figure 12. This looks like the representative figure of this study. I strongly would like to recommend you to use "the following view" here to show the vortical structures in each sequence. With the same wing attitude in 3d, the change in the vortices will be much more visible. The pivot point and stroke direction also should be addressed.
17) Page 14, Figure-of-merit. The figure-of-merit had been originally developed to see the efficiency of the helicopter rotors, and it has to be less than 1 due to its theoretical definition. This is because it is defined as the ratio between the ideal power for lift over the actual power for lift, and the actual power always accompanies the loss, i.e., actual power is always larger than the ideal one. In this study, the loss is found in the thrust direction, implying that the FM has to be much less than that of a helicopter or the values in Table 2. Here, however, I would like to rather recommend to use the lift-to-power ratio instead of FM. The authors can easily find out references that used the lift-to-power ratio to show efficiency; there is a lot.
18) Page 15, line 3. ARM here should be revised with SRM.
19) Conclusion. the expression '3d' looks weird as the 3d is the real world; I think if the authors used two-dimensional simulation then it needs to explain but it is not. The title also should be revised with considering this idea.
20) Reference [46]. (2001) should be removed.
21) Reference from [49] to [52]. Revise those.
Round 2
Reviewer 2 Report
Responding to the comments and concerns, the authors properly revised the manuscript; thank you very much.
I thus recommend the revised manuscript to be published in Fluids.